# Tumor-Infiltrating B Cells and Tissue-Resident Memory T Cells as Prognostic Indicators in Brain Metastases Derived from Gastrointestinal Cancers

**DOI:** 10.3390/cancers16223765

**Published:** 2024-11-08

**Authors:** Masasuke Ohno, Shunichiro Kuramitsu, Kimihiro Yamashita, Toru Nagasaka, Shoichi Haimoto, Mitsugu Fujita

**Affiliations:** 1Department of Neurosurgery, Aichi Cancer Center, Nagoya 464-8681, Japan; 2Department of Neurosurgery, Nagoya Medical Center, Nagoya 460-0001, Japan; 3Division of Gastrointestinal Surgery, Department of Surgery, Graduate School of Medicine, Kobe University, Kobe 650-0017, Japan; 4Association of Medical Artificial Intelligence Curation, Nagoya 460-0008, Japan; 5Center for Medical Education and Clinical Training, Kindai University Faculty of Medicine, Osaka-Sayama 589-8511, Japan

**Keywords:** brain metastases, gastrointestinal cancers, tumor-infiltrating B cells, tissue-resident memory T cells, prognostic indicators, tumor immune microenvironment, tertiary lymphoid structures

## Abstract

This study identified tumor-infiltrating B cells (TIBs) and tissue-resident memory T cells (TRMs) as potential prognostic indicators in brain metastases (BMs) derived from gastrointestinal (GI) cancers. Higher densities of TIBs and TRMs in the BM tissues significantly correlated with improved overall survival after BM diagnosis. These findings suggest that quantifying TIB and TRM levels in surgically resected BM samples could provide valuable prognostic information to guide treatment decisions and follow-up strategies for patients with this lethal condition. Additionally, we revealed distinct spatial distributions and characteristics of these lymphocyte subsets, which advanced our understanding of the BM immune microenvironment. Further studies with larger cohorts are needed to validate these findings and explore their potential therapeutic implications.

## 1. Introduction

Brain metastases (BMs), significant consequences of cancers, markedly impact prognosis and quality of life [1,2]. Although gastrointestinal (GI) cancers account for only 5–6% of primary cancers in patients with BMs [3], the prognosis of these patients is worse than those with BMs derived from other types of cancers [4]. This poor prognosis stems from an enhanced resistance to the current BM treatments, rapid intracranial progression, as well as a greater incidence of multiorgan metastases in GI cancers compared with other cancers [3]. Unlike non-small cell lung cancer, where genetic alterations such as EGFR mutations and ALK rearrangements serve as significant prognostic indicators and therapeutic targets for BMs [5], equivalent prognostic indicators remain unidentified for GI cancer-derived BMs (GIBMs) [4]. The urgent need for the identification of reliable prognostic indicators is essential to advance treatment efficacy for this lethal disease [6].

As the field of cancer immunotherapy advances, with immune checkpoint inhibitors (ICIs) and chimeric antigen receptor T (CAR-T) cell therapy making significant breakthroughs in cancer treatment, the tumor immune microenvironment (TIME) has emerged as a focal point of research. While the role of tumor-infiltrating T cells has been thoroughly investigated, that of tumor-infiltrating B cells (TIBs) has been relatively understudied. Recent studies have unveiled that TIBs also play significant roles in antitumor immunity [7]. They form ectopic lymph node-like structures known as tertiary lymphoid structures (TLSs) in conjunction with tumor-infiltrating lymphocytes (TILs) in the TIME. These TLSs, associated with improved outcomes in several cancer types [8], have rarely been reported in brain tumors despite their importance in antitumor immunity.

Tissue-resident memory T cells (TRMs) are a distinct subset of non-circulating T cells that persist in peripheral tissues, where they provide rapid and robust local immune responses to infections or malignancies [9]. Evidence suggests that the TRMs recruit B cells to the primary tumor sites [10,11,12,13], suggesting a potential role of TRMs in facilitating TIB accumulation within TIME.

To investigate the roles of TIBs and TRMs in GIBMs, we conducted analyses on surgically resected BM tissues. By analyzing both clinical parameters and pathological specimens of these cases, we aimed to clarify the prognostic significance of these cell types. The findings of this study may provide crucial insights for the development of novel prognostic indicators and immunotherapeutic strategies specifically tailored for patients with GIBMs.

## 2. Materials and Methods

### 2.1. Patients

The study protocol received approval from the Institutional Review Board at Aichi Cancer Center in Nagoya, Japan (approval number: IR041168); the requirement for written informed consent was waived due to the retrospective design of this study. Information about this study was made publicly available on our center’s website. Patient confidentiality was rigorously protected according to ethical guidelines. We performed retrospective histopathological analyses of archived formalin-fixed, paraffin-embedded surgical samples from 13 patients diagnosed with GIBMs who had undergone craniotomy at our institution between April 2018 and December 2022. Institutional pathologists conducted the histopathological examination and diagnosis. Clinical data, including patient demographics, tumor characteristics, and treatment histories, were extracted from medical records and analyzed along with the histopathological results.

### 2.2. Tissue Immunohistochemical Staining

The procedures used in this study have been previously described [14]. Briefly, tissue sections of 4-μm thickness underwent the following staining process. These sections were initially deparaffinized with xylene and rehydrated through a series of ethanol dilutions. Antigen retrieval was performed by heat-induced epitope retrieval. Subsequently, sections were either stained with HE or subject to immunostaining using the following primary monoclonal antibodies (mAbs): an anti-CD3 mouse mAb (clone PS1, Cat# 713241, Nichirei, Tokyo, Japan), an anti-CD4 rabbit mAb (clone EPR6855, Cat# ab133616, abcam, Cambridge, UK), an anti-CD20 mouse mAb (clone L26, Cat# 718471, Nichirei), an anti-BCL6 mouse mAb (clone LN22, Cat# 718181, Nichirei), an anti-CD8 mouse mAb (clone C8/144B, Cat# M710301-2, DAKO, Glostrup, Denmark), an anti-CD103 rabbit mAb (clone EPR4166(2), Cat# ab129202, abcam), an anti-PNAd rat mAb (clone MECA-79, Cat# NB100-77673, Novus biologicals, Centennial, CO, USA), and an anti-CD21 rabbit mAb (clone EP3093, Cat# ab75985, abcam). Dual staining for CD8 and CD103 utilized Histofine Simple Stain AP (M) (Cat# 414241, Nichirei) and Histofine Simple Stain MAX-PO (R) (Cat# 714341, Nichirei) as secondary antibody complexes. Chromogenic development was achieved using First Red II substrate kit (Cat# 415261, Nichirei) and HistoGreen substrate kit (Cosmo Bio, Cat# AYS E109, Tokyo, Japan). Triple staining for CD4, CD20, and BCL6 was performed in a multi-step process. First, CD20 detection was performed using Histofine Simple Stain MAX-PO (M) (Cat# 724132, Nichirei) and DAB substrate kit (Cat# 725191, Nichirei). This was followed by the hybridization of CD4 and BCL6 with Histofine Simple Stain MAX-PO (M) (Cat# 724132, Nichirei) and Histofine Simple Stain AP (R) (Cat# 414251, Nichirei), and chromogenic detection using First Red II substrate kit and HistoGreen substrate kit, respectively. CD3, PNAd, and CD21 were each chromogenized with the DAB substrate kit alone. Nuclei received contrast stained with Hematoxylin. Archived tonsil tissues were utilized as controls for the assays.

### 2.3. Histopathological Evaluation

The following procedure was performed blinded to patient clinical data. First, the stained tissue sections were digitally scanned using a NanoZoomer-SQ whole slide imaging system (Hamamatsu Photonics, Hamamatsu, Japan) with a 20 × 0.75 NA objective lens, according to standardized protocols. These images were subsequently processed using QuPath software (version 0.5.0) [15]. Tumor regions within each section were delineated using the “create threshold” tool. Then, each immune cell was identified by the “positive cell detection” function after the manual adjustment of color thresholds (color deconvolution). Immune cell density was quantified as the number of cells per mm^2^ within the tumor regions.

To quantify the spatial distributions of immune cells surrounding tumor vessels, we focused on cases presenting an adequate number of both TIBs and TRMs within the perivascular tumor stroma. For each selected case, five high-magnification regions of interest (ROIs) were randomly selected where immune cells clustered around blood vessels. These regions extended 500 μm from the vascular endothelium, capturing an equal proportion of tumor stroma and tumor epithelium. The proximity of individual immune cells to the closest vascular endothelium was measured using the “Distance to 2D annotations” feature under the “Spatial analysis” function [16]. Distance data were calculated for each subset of immune cells.

### 2.4. Statistical Analysis

Statistical analyses were performed using EZR software (version 1.64), as previously described [17,18]. Kruskal-Wallis test with Steel-Dwass’s post-hoc test was used for comparisons across multiple groups. Spearman’s rank correlation test was used to analyze the correlations between clinical parameters and TIL densities. Log-rank tests with Kaplan–Meier survival curves were used to analyze overall survivals (OS), where the patients were stratified into two groups (high and low groups) based on the median values of TIL subsets (TIBs, CD4^+^ T cells, non-TRMs, and TRMs). Statistical significance was defined as *p* < 0.05.

## 3. Results

### 3.1. Characteristics of Patients with Brain Metastasis Derived from GI Cancers

We examined the clinicopathological characteristics of the 13 patients with GIBMs who underwent surgical resection (Table 1). The histologies of primary lesions within this cohort included colorectal (*n* = 5, 38%), esophagus (*n* = 5, 38%), and stomach (*n* = 3, 23%). Among these patients, 9 patients (*n* = 9, 69%) presented with a single BM. Also, 10 patients (77%) had extracranial metastases. Prior to BM resections, 12 patients (92%) had received chemotherapy, while three patients (23%) and two patients (15%) had received radiation therapy and ICI therapy, respectively. We observed no significance among these data.

### 3.2. TIBs, CD4^+^T Cells, and TRMs Accumulate in BMs

Then, we focused on the identification of TLSs and their clinical significance within GIBMs. TLSs are structures typically located within the tumor stroma and peritumoral areas of the primary solid cancers [19]. Therefore, we first visually distinguished the tumor epithelium, composed of the malignant cells, and the tumor stroma, composed of the blood vessels or infiltrating immune cells, in the GIBM tissues. Prior to analyzing TIL subsets, we performed CD3 immunostaining to visualize the overall T cell distribution. The distribution of CD3^+^ T cells in serial tissue sections (Appendix A) corresponded to the combined distribution patterns of CD4^+^ and CD8^+^ cells (Appendix A, respectively). We then focused on analyzing the distribution of CD20^+^ TIBs and CD4^+^ T cells, recognized as the primary components of TLSs [19]. TIBs (Figure 1A, brown) were predominantly found around the blood vessels (white arrowhead) within the tumor stroma (yellow dotted line). Conversely, CD4^+^ T cells (Figure 1A, red) were more widely distributed, extending from the meninges (white arrows) to the tumor stroma (yellow dotted line), yet similarly clustered in the regions where TIBs were clustered (blue dotted line). To determine whether these lymphocyte clusters would share the characteristics of TLSs previously reported, we evaluated the expression of BCL6, an indicator of germinal centers (GCs) [19]. Although BCL6^+^ cells were sporadically observed, typical GC-like clusters of BCL6^+^ TIBs were absent (Figure 1B, black arrowheads). We also examined the expressions of CD21 and PNAd, indicators for follicular dendritic cells (FDCs) and high endothelial venules (HEVs), respectively, key components of TLSs [20]. Nevertheless, similar to BCL6 expression, the lymphocyte clusters in BMs lacked CD21^+^ and PNAd^+^ cells (Appendix A).

Next, we examined the distribution of CD8^+^CD103^+^ TRMs in the GIBMs, given their role in facilitating the recruitment of TIBs and the development of TLSs by chemokine secretion [10,11,12,13]. CD20^+^ TIBs and CD4^+^ T cells were predominantly located within the tumor stroma (Figure 1A, yellow dotted line). Conversely, CD8^+^CD103^-^ T cells (non-TRMs: Figure 1C, red) and CD8^+^CD103^+^ TRMs (Figure 1C, black arrows), which appear black due to the overlay of red (CD8) and green (CD103) signals, were primarily distributed within tumor stroma, extending into the adjacent tumor epithelium (Figure 1C, red dotted line). High-magnification views revealed that the TRMs penetrated the tumor epithelium more extensively than their non-TRMs counterparts (Figure 1D). These data suggest the presence of TLS-like lymphocyte clusters within the tumor stroma, composed of CD20^+^ TIBs and CD4^+^ T cells, yet lacking the key TLS characteristics such as BCL6^+^ GCs, CD21^+^ and PNAd^+^ cells (Appendix A). Furthermore, the distinct distribution patterns of TRMs and non-TRMs concerning the tumor epithelium and stroma suggest the potential roles of TRMs in the TIME of GIBMs.

### 3.3. TIBs and TRMs Manifest Differential Distributions Within BM Tissues

To further evaluate the spatial characteristics of TIBs and TRMs within the GIBM tissues, we analyzed their proximity to tumor-associated blood vessels. This analysis focused on four cases (Cases 2, 3, 9, and 11) noted for their significant presence of both cell types in the perivascular regions of the tumor stroma (Figure 2A,B, red circle). The distance between these immune cells and nearby blood vessels was independently measured and analyzed. TIBs were located significantly closer to blood vessels than non-TRMs (*p* < 0.001; Figure 2C), and the non-TRMs were located significantly closer to blood vessels than TRMs (*p* < 0.001). These data suggest a preferential distribution of TIBs within tumor stroma, while TRMs would exhibit the ability to penetrate deeper into the TIME and potentially maintain their presence longer than both TIBs or non-TRMs.

### 3.4. TIB Density in BM Tissues Correlates with OS Including GPA

Using the methods above, we quantified the densities of the TIL subsets (TIBs, CD4^+^ T cells, TRMs, and non-TRMs) within the GIBM tissues across all cases (Table 2). Then, we conducted comprehensive analyses of the correlations between clinical parameters and each TIL density. The patients’ age, BM counts, and presence of extracranial metastatic lesions did not correlate with any TIL density. However, the KPS correlated with the densities of TIB and CD4^+^ T cells (Appendix A). We then focused on the OS. First, we used Graded Prognostic Assessment (GPA), a robust prognostic tool for BMs [6], to estimate the predictive value of TIL densities for OS. We observed a significant correlation between TIB density and the prognostic prediction of GPA (*p* = 0.049; Figure 3A).

Next, we examined the correlations between the densities of TIL subsets and OS across various stages. While the TIB density did not correlate with OS after primary lesion diagnosis (*p* = 0.654; Figure 3B), it significantly correlated with OS after BM diagnosis (*p* = 0.029; Figure 3C) or BM surgery (*p* = 0.041; Figure 3D). Other TIL densities (CD4^+^ T cells, TRMs, and non-TRMs) showed no significant correlation with these survival data (Appendix A). These data suggest the potential of the TIB density as a reliable prognostic indicator for the survival of patients with GIBMs at various disease stages.

### 3.5. Densities of TIBs and TRMs Correlate with OS After BM Diagnosis

As we observed the positive correlation between TIB density and the mean survival, including the estimated survival by GPA and OS after BM diagnosis or surgery (Figure 3), we processed detailed survival analyses. The patients were stratified into two groups based on the median density of each TIL subset (TIBs, CD4^+^ T cells, TRMs, and non-TRMs). No significant impact on OS after primary lesion diagnosis or that after BM surgery was observed across any TIL subsets (Appendix A). Subsequent analyses focused on the OS after BM diagnosis revealed that the densities of the TIBs (*p* = 0.0351; Figure 4A) and the TRMs (*p* = 0.0254; Figure 4D) significantly correlated with the survival. Conversely, CD4^+^ T cells (*p* = 0.259; Figure 4B) and non-TRMs (*p* = 0.124; Figure 4C) did not correlate. These data highlight the potential of TIB density as a reliable prognostic indicator and suggest that TRM density, while not as robust, could also serve as a prognostic indicator for patients with GIBMs.

## 4. Discussion

We examined the distribution of lymphocytes in GIBMs and identified clusters of TIBs and CD4^+^ T cells in the tumor stroma that exhibited distinct characteristics from those of TLSs in primary GI cancers (Figure 1). TRMs were extensively distributed across the tumor stroma and epithelium, whereas TIBs were predominantly located in perivascular regions (Figure 2). We then demonstrated positive correlations between the TIB density and OS after BM diagnosis as well as BM surgery using both comprehensive and survival analyses (Figure 3 and Figure 4). These data suggest that TIB density has a promising potential to serve as a reliable prognostic indicator for predicting survival outcomes in patients with GIBMs.

In this study, we identified lymphocyte clusters composed of TIBs and CD4^+^ T cells predominantly located within the perivascular niche along tumor margins or stroma of GIBMs, which we interpreted as TLSs (Figure 1 and Figure 2). They develop at the sites of chronic inflammation or tumors, exhibit various cellular components, and range from nascent to fully mature structures [21]. Mature TLSs exhibit well-organized lymphoid architecture, characterized by BCL6^+^ GCs, discrete B cell zones with FDCs, adjacent T cell zones, and HEVs [20]. Conversely, immature TLSs lack these defining characteristics and structures as they are still in the developmental phase. To date, TLSs in brain tumors have been reported in only two studies, both focusing on glioblastoma [22,23], where the described TLSs resembled the immature TLSs we observed in our cases. It has been reported that TIBs predominantly reside within TLSs [19] and that BMs contain more TIBs than glioblastomas [24]. These findings predict that TLSs may develop within GIBMs more than within glioblastomas. Nevertheless, only immature TLSs were observed within the GIBMs (Figure 1), similar to glioblastomas. These data suggest that the development of mature TLSs in brain tumors may be challenging due to the unique immunological and anatomical features of the brain [22].

Mature TLSs are known to correlate with improved prognosis and responses to immunotherapy in various cancers [8], yet the role of immature TLSs in brain tumors, including GIBMs, remains unclear. We observed the positive correlation between TIB density and GPA-predicted median survival in the patients with GIBMs (Figure 3A), emphasizing the need for careful interpretation of these findings. The GPA, a prognostic tool for patients with BMs, incorporates critical factors such as age, KPS, the presence of extracranial metastases, and BM counts [6], but it does not account for immunological factors within the BMs. However, GPA elements such as primary cancer types and the patient’s overall physical condition may indirectly affect the immune activities within BMs. For instance, ICI-based systemic treatments have been suggested to affect the TIME of BMs and, therefore, patient outcomes [25]. Our findings highlight only a component of these complex interactions, suggesting the necessity for future studies to clarify the immunological interplay between TIBs and prognosis in patients with GIBMs.

Survival analyses revealed that higher densities of TIBs (Figure 4A) and TRMs (Figure 4D) significantly correlated with longer OSs after BM diagnosis in the patients with GIBMs. This finding suggests the potential of these cell types as robust prognostic indicators. Historically, the exploration of immune cells within tumors for prognostic has begun with the study of TILs as a whole [26]. Subsequent studies have identified CD8^+^ cytotoxic T lymphocytes (CTLs), known for their primary role in antitumor immunity, as a promising prognostic indicator [26]. Despite the ability of CTLs to penetrate tumor core and their prevalence within tumor tissue (Figure 1C,D), the focus on TIBs has been limited due to their perceived indirect roles in antitumor immunity [19] and their sparse distributions within the tumor tissues as observed in this study (Figure 1A,B). However, recent studies have revealed that TIBs play crucial roles in antitumor immunity as essential components of TLSs and lymphomyeloid aggregates (LMAs; Figure 1A,B). Within these structures, TIBs mediate multiple antitumor immune responses, including tumor-specific antibody production [7], antibody-dependent cellular cytotoxicity (ADCC) [7], antibody-dependent cell-mediated phagocytosis (ADCP) [7], tumor antigen presentation to T cells [19], and cytokine secretion required for these responses [27]. The presence of TLSs and LMAs has been associated with favorable prognosis and improved immunotherapy responses in several cancers, including non-small cell lung cancer, colorectal cancer, and breast cancer [19,27]. Consistent with these previous studies, the results of this study showed that the TIBs significantly correlate with improved survival after BM diagnosis (Figure 3 and Figure 4A). In this regard, to further advance the histopathological evaluation of TIBs, we are in the process of implementing a deep learning-based image analysis platform “Cu-Cyto” [28] in addition to the “QuPath” platform.

Our attention was directed towards the detailed characteristics of TRMs within BM tissues. While both CD69 and CD103 are established markers of TRMs, we focused on CD103 because it is a more stable and specific marker of long-term tissue residence in tumor tissues [29]. In contrast to CD69, which can be transiently upregulated upon T cell activation and serves as an early activation marker, CD103 expression indicates established tissue residence through its interaction with E-cadherin [29]. Furthermore, CD103^+^ TRMs are associated with enhanced cytotoxic function, which correlates with improved clinical outcomes in several cancers [29]. As a result, TRMs were identified to have a distinct distribution, penetrating deeper into the tumor epithelium than non-TRMs (Figure 1 and Figure 2). TIBs highly express the chemokine receptor CXCR5 [29] while TRMs secrete its ligand CXCL13 chemokine [10,11], suggesting a mechanism through which TRMs would recruit the TIBs towards the TIME of BMs. This hypothesis is supported by the observation that TRMs are located deep within the tumor mass, whereas TIBs are primarily found in the perivascular areas (Figure 1 and Figure 2). Further studies are needed to clarify the distinct roles and interactions between TIBs and TRMs in the context of brain tumors.

Limitations of this study include a small sample size, the exclusive inclusion of surgically treated cases, the inclusion of various pathological types of GI cancers as primary lesions, and the lack of correlation with the histologic evaluation of these primary lesions. The additional analysis revealed that Case 9 showed elevated lymphocyte densities across all subsets, particularly for TIBs, with its exclusion affecting the statistical significance. The retention of this case, despite its extreme values, was justified by its potential biological significance and limited sample size considerations. Several factors may explain the elevated values of Case 9, including patient-specific characteristics, tumor heterogeneity, and methodological variations. In addition, Case 9’s preoperative radiation therapy may have contributed through bystander and abscopal effects, potentially triggering anti-tumor immunity via tumor antigen release [30]. However, our small sample size limited the definitive determination of whether this represents a true biological phenomenon or a statistical anomaly. This point of discussion emphasizes the preliminary nature of our findings and the need for larger validation studies to better understand the TIB intensity across larger patient populations.

## 5. Conclusions

In conclusion, we identified TLS-like structures composed of TIBs and TRMs in the GIBMs. Remarkably, increased densities of TIBs and TRMs correlated with enhanced OS after BM diagnosis, suggesting their potential as reliable prognostic indicators for this lethal disease. These findings provide valuable insights into the development of novel immunotherapeutic strategies tailored for patients with BMs.

## Figures and Tables

**Figure 1 cancers-16-03765-f001:**
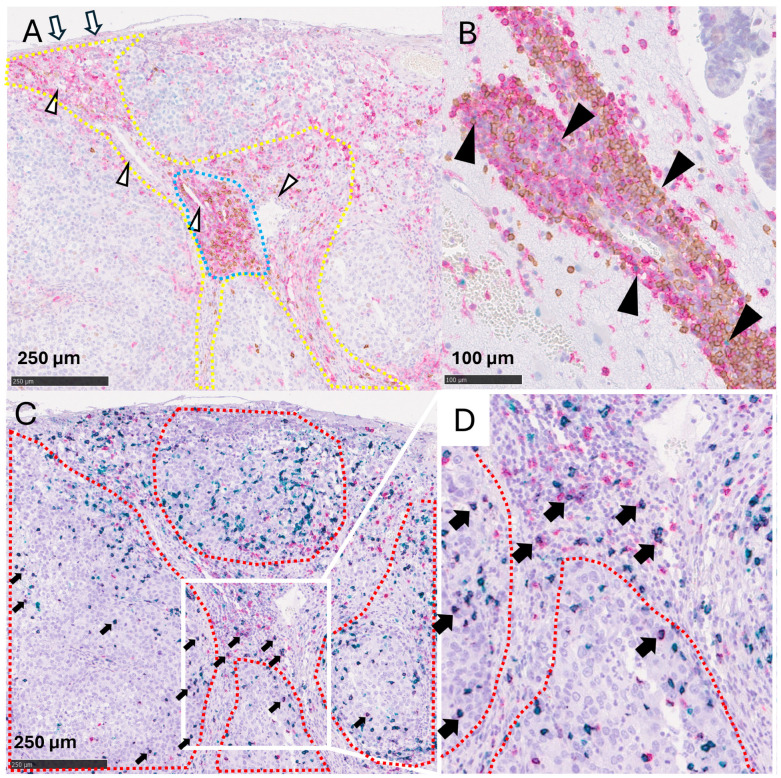
Tumor-infiltrating B cells (TIBs), CD4^+^ T cells, and tissue-resident CD8^+^ T cells (TRMs) accumulate in brain metastases (BMs) derived from gastrointestinal (GI) cancers. Representative histopathological images of BMs from GI cancers. (**A**,**B**): CD20 (brown), CD4 (red), and BCL6 (green) were simultaneously stained. (**A**): CD20^+^ B cells and CD4^+^ T cells are confined within the tumor stroma (yellow dotted line) and clustered together (blue dotted line). White arrowheads indicate blood vessels within the tumor stroma. Tumor-meninges boundary (white arrow). Scale bars = 250 μm. (**B**): CD20^+^ B cells and CD4^+^ T cells expressing BCL6 (black arrowheads). Scale bar = 100 μm. (**C**): Dual immunohistochemistry for CD8 and CD103 on serial tissue sections of the sample shown in Panel A. CD8^+^ T cells and CD8^+^CD103^+^ T cells strongly infiltrate the tumor epithelium (red dotted line). CD8 (red) and CD103 (green); CD8^+^CD103^+^ T cells are stained black. Scale bars = 250 μm. (**D**): Magnified view of (**C**). CD8^+^CD103^+^ T cells infiltrate deeper into the tumor epithelium compared to CD8^+^ T cells that do not express CD103. Scale bars = 100 μm.

**Figure 2 cancers-16-03765-f002:**
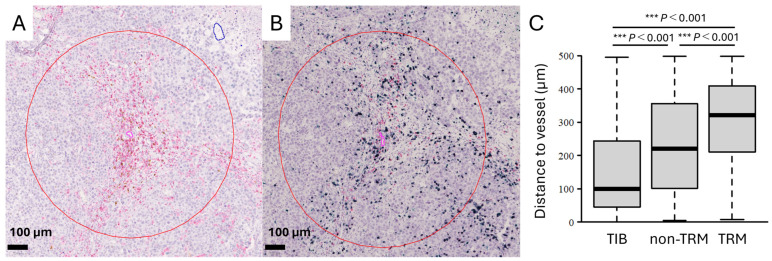
TIBs and TRMs manifest differential distributions within BM tissues. Representative immunohistochemistry of consecutive tissue sections showing (**A**) CD20^+^B-cells (brown) and (**B**) CD8^+^CD103^-^T-cells (red) and CD8^+^CD103^+^T-cells (black) residing in the perivascular areas of tumor blood vessels (pink). The 500 μm distance from tumor blood vessels is indicated by the red line. Scale bar, 100 μm. (C) Box plot showing the distance from blood vessels to TIBs, non-TRMs, and TRMs. The distances were measured in five randomly selected regions of interest from four cases with sufficient numbers of all three cell types. ***, *p* < 0.001.

**Figure 3 cancers-16-03765-f003:**
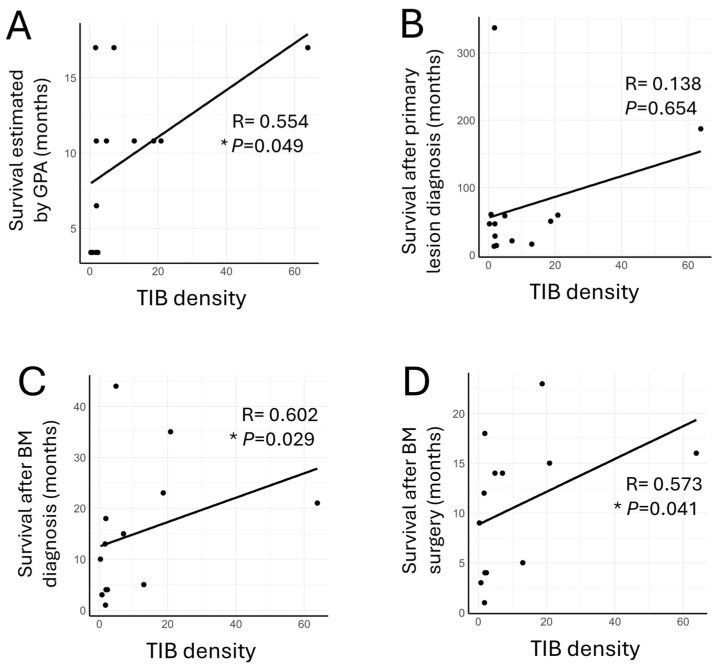
TIB density in BM tissues correlates with OS including GPA. Spearman’s rank correlation coefficient analyses between TIB density and OS after (**A**) the estimated survival by the Graded Prognostic Assessment (GPA), (**B**) primary lesion diagnosis, (**C**) BM diagnosis, and (**D**) BM surgery. *, *p* < 0.05.

**Figure 4 cancers-16-03765-f004:**
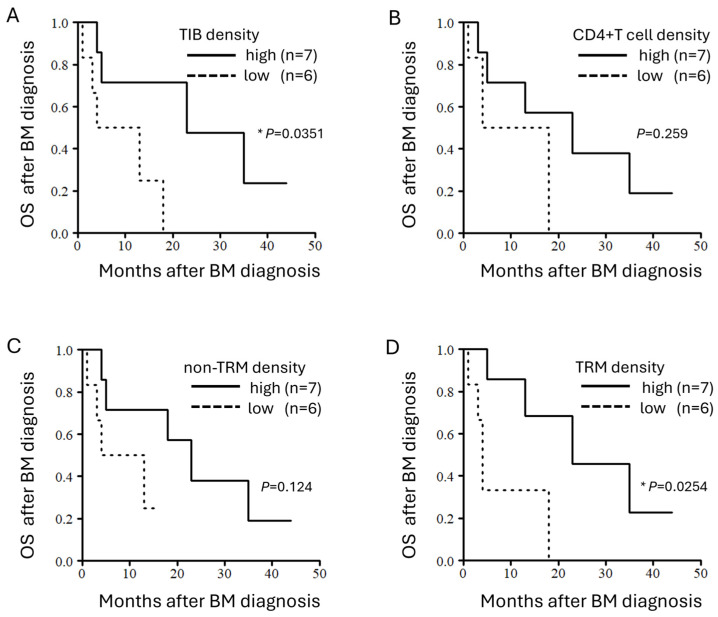
Densities of TIBs and TRMs correlate with OS after BM diagnosis. Kaplan–Meier curves for OS after BM diagnosis stratified by the density (high vs. low) of (**A**) TIBs, (**B**) CD4^+^ T cells, (**C**) non-TRMs, and (**D**) TRMs. The patients were dichotomized into high and low groups based on the median value of each TIL subset density. *, *p* < 0.05.

**Table 1 cancers-16-03765-t001:** Summary of the clinical characteristics of patients with brain metastasis (BMs) derived from gastrointestinal (GI) cancers.

Patient Characteristics	Value
No. of patients	13
Age at diagnosis of BM, years, median (range)	67 (35–79)
Sex, No. (%)	
Female	3 (23)
Male	10 (77)
BM status, No. (%)	
Single	9 (69)
Multiple	4 (31)
KPS, median (range)	80 (40–90)
Extra-cranial metastasis, No. (%)	
No	3 (23)
Yes	10 (77)
MST predicted by the GPA, months, (range)	10.8 (3.4–17)
Histology of the primary lesions, No. (%)	
Colorectal	5 (38)
Esophageal	5 (38)
Gastric	3 (23)
Preoperative steroid treatment, No. (%)	
No	6 (46)
Yes	7 (54)
Previous chemotherapy, No. (%)	
No	1 (8)
Yes	12 (92)
Previous radiation therapy, No. (%)	
No	10 (77)
Yes	3 (23)
Previous immune checkpoint inhibitor treatment, No. (%)	
No	11 (85)
Yes	2 (15)
Postoperative chemotherapy, No. (%)	
No	5 (38)
Yes	4 (31)
N/A *	4 (31)
Postoperative radiation therapy, No. (%)	
No	2 (15)
Yes	10 (77)
N/A *	1 (8)

BM: brain metastasis, KPS: Karnofsky Performance Status, MST: median survival time, GPA: Graded Prognostic Assessment. * N/A: Data not available for patients who are still under treatment or who have not died.

**Table 2 cancers-16-03765-t002:** Details of the patient characteristics including immune cell densities in the BM tissues.

							Survival After Clinical Time Point (Months)				Immune Cell Density (/mm^2^)			
Case	Age	KPS	BM Counts	Extra-Cranial Metastasis	Type of Primary Tumor	Current Status	Estimated by GPA	Primary Legion Diagnosis	BM Diagnosis	BM Surgery	TIB	CD4^+^ T	TRM	Non-TRM
1	42	80	1	Yes	Colorectal	death	10.8	46	18	18	1.89	6.85	0.00	4.83
2	53	80	2	Yes	Esophageal	death	10.8	16	5	5	13.00	133.27	90.97	404.66
3	67	80	1	Yes	Esophageal	death	10.8	59	35	15	20.85	306.35	15.16	61.82
4	35	90	1	Yes	Colorectal	death	17	13	13	12	1.68	199.90	0.45	2.42
5	46	40	3	Yes	Colorectal	death	3.4	60	3	3	0.74	82.44	0.00	1.55
6	56	50	10	No	Esophageal	death	3.4	14	4	4	2.37	1.24	0.15	10.79
7	72	90	1	Yes	Gastric	death	10.8	50	23	23	18.71	289.27	0.32	8.66
8	77	50	1	Yes	Gastric	death	6.5	28	4	4	1.97	62.03	0.01	1.30
9	53	90	1	Yes	Colorectal	alive	17	187	21	16	63.81	956.11	4.20	57.64
10	68	80	1	No	Esophageal	alive	17	21	15	14	7.06	12.26	0.00	0.67
11	68	80	1	Yes	Gastric	alive	10.8	58	44	14	4.88	186.03	1.19	14.84
12	79	50	2	No	Colorectal	death	3.4	337	1	1	1.80	25.80	0.00	0.25
13	73	50	1	Yes	Esophageal	alive	3.4	46	10	9	0.30	42.77	1.11	0.42

KPS: Karnofsky Performance Status, BM: brain metastasis, GPA: Graded Prognostic Assessment.

## Data Availability

The datasets generated during and/or analyzed during the current study are available from the corresponding author upon reasonable request.

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
