# Peer review of "Tumor-Infiltrating B Cells and Tissue-Resident Memory T Cells as Prognostic Indicators in Brain Metastases Derived from Gastrointestinal Cancers"

_cancers, 2024, doi:10.3390/cancers16223765_

Round 1
Reviewer 1 Report
Comments and Suggestions for Authors
Summary- This study identified tumor-infiltrating B cells (TIBs) and tissue-resident memory T cells (TRMs) as potential prognostic indicators in brain metastases (BMs) derived from gastrointestinal (GI) cancers. Improved overall survival following BM diagnosis was substantially linked with higher densities of TIBs and TRMs in the BM tissues. These results imply that measuring TIB and TRM levels in surgically removed bone marrow samples may offer insightful prognostic data to inform treatment choices and post-mortem plans for individuals suffering from this fatal illness. Moreover, the unique spatial distributions and functional traits of TRMs and TIBs reported in this work offer vital information for creating innovative immunotherapies.
1. What is the function and significance of TIB in brain metastases (BMs) derived from gastrointestinal (GI) cancers?
2. IHC images can be further improved in quality.
3. CD103 is expressed only by a subset of tissue memory CD8+ and not by CD4+ T cells. Did author check for the expression of CD69?
Author Response
Comment 1: “What is the function and significance of TIBs in brain metastases (BMs)
derived from gastrointestinal (GI) cancers?”
As the reviewers pointed out, the functions and significance of TIBs in GI-derived BMs were described in a scattered but not comprehensive manner in the original manuscript. They can be summarized as follows:
1. Formation of TLSs (Figures 1A and 1B) [7].
2. Antibody production and ADCC for tumor cell killing [7].
3. ADCP and antigen presentation to T cells [7, 19].
4. Cytokine secretion to modulate immune responses [27].
5. Prognostic indicator for survival in cancers (Figures 3 and 4A) [19, 27].
Accordingly, we have added the following statement in the Discussion section to provide a summary of the functions and significance of TIBs in GI-derived BMs:
However, recent studies have revealed that TIBs play crucial roles in antitumor immunity as essential components of TLSs and lymphomyeloid aggregates (LMAs; Figure 1A, B). Within these structures, TIBs mediate multiple antitumor immune responses, including tumor-specific antibody production [7], antibody-dependent cellular cytotoxicity (ADCC) [7], antibody-dependent cell-mediated phagocytosis (ADCP) [7], tumor antigen presentation to T cells [19], and cytokine secretion required for these responses [27]. The presence of TLSs and LMAs has been associated with favorable prognosis and improved immuno-therapy responses in several cancers, including non-small cell lung cancer, colorectal cancer, and breast cancer [19,27]. Consistent with these previous studies, the results of this study showed that the TIBs significantly correlate with improved survival after BM diagnosis (Figure 3, 4A).
Comment 2: “IHC images can be further improved in quality.”
In fact, in the original submission, we had already provided high-resolution image files that met publication quality standards in addition to the figures embedded in the manuscript. We would like the reviewer to refer the high-resolution figures on the web system.
Comment 3: “CD103 is expressed only by a subset of tissue memory CD8+ and not by CD4+ T cells. Did author check for the expression of CD69?”
In this study, we did not evaluate CD69 expression levels. While we acknowledge that CD69 is a widely used marker of TRM cells, we specifically focused on CD103 expression in our study because it is a more stable and specific marker of long-term tissue-residence [29]. In contrast to CD69, which can be transiently upregulated upon T cell activation and serves as an early activation marker, CD103 expression indicates established tissue-residence through its interaction with E-cadherin [29]. Furthermore, CD103+ TRMs are associated with enhanced cytotoxic function, which correlates with
improved clinical outcomes in several cancers [29], making CD103 a more appropriate marker for this study.
Based on the discussion above, we have added the following statement in the Discussion section.
Our attention was directed towards the detailed characteristics of TRMs within BM tissues. While both CD69 and CD103 are established markers of TRMs, we focused on CD103 because it is a more stable and specific marker of long-term tissueresidence in tumor tissues [29]. In contrast to CD69, which can be transiently upregulated upon T cell activation and serves as an early activation marker, CD103
expression indicates established tissue-residence through its interaction with Ecadherin [29]. Furthermore, CD103+ TRMs are associated with enhanced cytotoxic function, which correlates with improved clinical outcomes in several cancers [29] . As a result, TRMs were identified to have a distinct distribution, penetrating deeper into the tumor epithelium than non-TRMs (Figures 1 and 2). TIBs highly express the chemokine receptor CXCR5 [29] while TRMs secrete its ligand CXCL13 chemokine
[10,11], suggesting a mechanism through which TRMs would recruit the TIBs towards the TIME of BMs. This hypothesis is supported by the observation that TRMs are located deep within the tumor mass, whereas TIBs are primarily found in the perivascular areas (Figures 1 and 2). Further studies are needed to clarify the distinct roles and interactions between TIBs and TRMs in the context of brain tumor.
Reviewer 2 Report
Comments and Suggestions for Authors
In this manuscript, the authors investigated the spatial distributions and densities of Tumor-infiltrating B cells (TIBs) and tissue-resident memory T cells (TRMs) in brain metastases (BMs) tissues from 13 patients with gastrointestinal cancers who underwent surgical resection. The present study suggested that TIB and TRM densities in BM tissues could serve as reliable prognostic indicators for survival in patients with brain metastases derived from gastrointestinal cancers. I think that the topic and obtained results are interesting. However, I have some concerns in this manuscript.
Comments:
1. The authors have to add the information about catalog number of the antibodies used in this study.
2. In this study, the authors claimed that they detected CD4 T cells. However, it is known that CD4 is expressed on not only helper T cells but also other immune cells. I think that the authors should stain the CD3 which is absolute marker for T cells. How did the authors determine that CD4+ cells were CD4 T cells without staining CD3?
3. Figure 3: As the authors mentioned that limitations of this study include small sample size. I think this is very important issue. Although Figure 3 showed some statistically significant but weak difference (p value are almost 0.05), it might be due to an outlier (presumably case 9). Can the authors observe the significant differences when the data of case 9 are excluded? In addition, the authors have to discuss about the reason why the TIB intensity of case 9 was extremely different from that of the other cases.
4. There are some careless mistakes such as “mm2” in Line 124 and “Figures, in Tables and Schemes” in Line 263, etc.
Author Response
Response to Reviewer 2
Comment 1: “The authors have to add the information about catalog number of the antibodies used in this study.”
As suggested by the reviewer, we have added the following information about the catalog numbers of the antibodies used in this study:
Subsequently, sections were either stained with HE or subject to immunostaining using the following primary monoclonal antibodies (mAbs): an anti-CD3 mouse mAb (clone PS1, Cat# 713241, Nichirei, Tokyo, Japan) an an-ti-CD4 rabbit mAb (clone EPR6855, Cat# ab133616, abcam, Cambridge, UK), an an-ti-CD20 mouse mAb (clone L26, Cat# 718471, Nichirei), an anti-BCL6 mouse mAb (clone LN22,
Cat# 718181, Nichirei), an anti-CD8 mouse mAb (clone C8/144B, Cat# M710301-2, DAKO, Glostrup, Denmark), an anti-CD103 rabbit mAb (clone EPR4166(2), Cat# ab129202, abcam), an anti-PNAd rat mAb (clone MECA-79, Cat# NB100-77673, Novus biologicals, Centennial, CO, USA), and an anti-CD21 rabbit mAb (clone EP3093, Cat# ab75985, abcam). Dual staining for CD8 and CD103 utilized Histofine Simple Stain AP (M) (Cat# 414241, Nichirei) and Histofine Simple Stain MAX-PO (R) (Cat# 714341, Nichirei) as secondary antibody complexes. Chromogenic development was achieved using First Red II substrate kit (Cat# 415261, Nichirei) and HistoGreen substrate kit (Cosmo Bio, Cat# AYS E109, Tokyo,
Japan). Triple staining for CD4, CD20, and BCL6 was performed in a multistep process. First, CD20 detection was performed using Histofine Simple Stain MAXPO (M) (Cat# 724132, Nichirei) and DAB substrate kit (Cat# 725191, Nichirei). This was followed by the hybridization of CD4 and BCL6 with Histofine Simple Stain MAX-PO (M) (Cat# 724132, Nichirei) and Histofine Simple Stain AP (R) (Cat#
414251, Nichirei), and chromogenic detection using First Red II substrate kit and HistoGreen substrate kit, respectively.
Comment 2: “In this study, the authors claimed that they detected CD4 T cells. However, it is known that CD4 is expressed on not only helper T cells but also other immune cells. I think that the authors should stain the CD3 which is absolute marker for T cells. How did the authors determine that CD4+ cells were CD4 T cells without staining CD3?”
As we agreed with the reviewer, we have performed the analysis of CD3 expression levels (Figure S1A). Accordingly, we have added the following statements in the Methods, Results, and Discussion sections.
(Methods)
Subsequently, sections were either stained with HE or subject to immunostaining using the following primary monoclonal antibodies (mAbs): an anti-CD3 mouse mAb (clone PS1, Cat# 713241, Nichirei, Tokyo, Japan), CD3, PNAd, and CD21 were each chromogenized with the DAB substrate kit alone.
(Results)
Prior to analyzing TIL subsets, we performed CD3 immunostaining to visualize the overall T cell distribution. The distribution of CD3+ T cells in serial tissue sections (Figure S1A) corresponded to the combined distribution patterns of CD4+ and CD8+ cells (Figure S1B, C, respectively).
Comment 3. “Figure 3: As the authors mentioned that limitations of this study include small sample size. I think this is very important issue. Although Figure 3 showed some statistically significant but weak differences (p value are almost 0.05), it might be due to an outlier (presumably case 9). Can the authors observe the significant differences when the data of case 9 are excluded? In addition, the authors have to discuss about the reason why the TIB intensity of case 9 was extremely different from that of the other cases.”
As the reviewer pointed out, it is possible that Case 9 is an outlier. In fact, the statistical significance is lost if Case 9 is excluded. However, we decided to retain Case 9 for the following reasons. First, excluding this case would further reduce the already limited sample size (n=13) and compromise the statistical power. Second, Case 9 is a valuable lymphocyte-rich phenotype and is worth investigating rather than excluding. The high lymphocyte infiltration and favorable prognosis in Case 9 may be explained by two key factors. First, this case was one of only three that had received preoperative
radiotherapy, which could enhance immune cell recruitment through bystander and abscopal effects. Second, the patient was one of three cases with the highest GPA scores in our cohort, suggesting that the favorable prognostic factors included in the GPA may reflect good systemic conditions capable of maintaining proper immune function.
Based on the Reviewer’s comments and the discussion points above, we have added the following statement in the Discussion section.
The additional analysis revealed that Case 9 showed elevated lymphocyte densities across all subsets, particularly for TIBs, with its exclusion affecting the statistical significance. The retention of this case, despite its extreme values, was justified by its potential biological significance and limited sample size
considerations. Several factors may explain the elevated values of Case 9, including patient-specific characteristics, tumor heterogeneity, and methodological variations. In addition, Case 9's preoperative radiation therapy may have contributed through bystander and abscopal effects, potentially triggering
anti-tumor immunity via tumor antigen release [30]. However, our small sample size limited definitive determination of whether this represents a true biological phenomenon or a statistical anomaly. This point of discussion emphasizes the preliminary nature of our findings and the need for larger validation studies to better understand the TIB intensity across larger patient populations.
Comment 4: “There are some careless mistakes such as “mm2” in Line 124 and “Figures, in Tables and Schemes” in Line 263, etc.”
Based on the reviewer's comments, we have corrected all typos that we are aware of.
Below are the areas you pointed out, all of which have been corrected. We will attempt to correct all other typos whenever possible, with the assistance of the journal's editorial staff. We appreciate your attention to detail, as it helps us improve the quality of our paper.
1. We corrected "mm2" to "mm2" on Line 127.
2. We removed the unnecessary heading "Figures, Tables and Schemes" from Line 269.